# Respiratory Dysfunctions in Parkinson’s Disease Patients

**DOI:** 10.3390/brainsci11050595

**Published:** 2021-05-04

**Authors:** Any Docu Axelerad, Alina Zorina Stroe, Oana Cristina Arghir, Daniel Docu Axelerad, Anca Elena Gogu

**Affiliations:** 1Department of Neurology, General Medicine Faculty, Ovidius University, 900470 Constanta, Romania; docuaxi@yahoo.com; 2Department of Pneumology, Faculty of Medicine, Ovidius University of Constanta, 900470 Constanta, Romania; arghir_oana@yahoo.com; 3Department of Kinetotherapy, Brainaxy Clinic, 900628 Constanta, Romania; docuaxy@yahoo.com; 4Department of Neurology, Victor Babes University of Medicine and Pharmacy, 300041 Timisoara, Romania; agogu@yahoo.com

**Keywords:** Parkinson’s disease, respiratory dysfunction, pulmonary complications

## Abstract

Respiratory dysfunctions have been associated with Parkinson’s disease since the first observations of the disease in 1817. Patients with Parkinson’s disease frequently present respiratory disorders with obstructive ventilatory patterns and restrictive modifications, as well as limitations in respiratory volumes. In addition, respiratory impairments are observed due to the rigidity and kyphosis that Parkinson’s disease patients experience. Subsidiary pulmonary complications can also appear as side effects of medication. Silent aspiration can be the cause of pneumonia in Parkinson’s disease. Pulmonary dysfunction is one of the main factors that leads to the morbidity and mortality of patients with Parkinson’s disease. Here, we performed a narrative review of the literature and reviewed studies on dyspnea, lung volumes, respiratory muscle function, sleep breathing disorders, and subsidiary speech and swallow impairments related to pulmonary dysfunction in patients with Parkinson’s disease.

## 1. Introduction

Parkinson’s disease is defined as a progressive neurodegenerative disorder caused by the depletion of dopamine in the basal ganglia, which is primarily involved in motor control. Parkinson’s disease is viewed as a multisystem disease that affects systemic dopaminergic neurons. The motor symptoms include rigidity, tremor, bradykinesia, akinesia, postural instability, and gait disturbances [1]. 

The cardinal symptoms of the disease consist of the triad of tremor, rigidity, and bradykinesia, with bradykinesia being the primary symptom that has an impact on the daily activity of patients. Additionally, subsidiary dysfunctions include the following: dysfunctions in the musculoskeletal system such as flexed positions and contractures as well as dysfunctions in the cardiopulmonary system which can aggravate the primary symptoms [2]. 

Studies have reported a possible etiology of Parkinson’s disease to be the primary anatomical neurodegenerative involvement of structures in the medulla oblongata known to control respiratory depth and rate [3,4]. 

This could also be the etiology of the respiratory dysfunctions in the initial stages of the disease related to lung volumes and ventilatory capacity. 

The risk for Parkinson’s disease patients is often an increased risk of morbidity and mortality due to dysfunctions in the respiratory system including obstructive and restrictive ventilatory defect patterns and upper airway and intercostal muscle problems, which cause a reduction in quality of life [5]. Furthermore, a significant number of Parkinson’s disease patients die due to pneumonia as compared with the general population [6]. 

In his 1817 essay “An Essay on the Shaking Palsy”, James Parkinson made the first observation, “he fetched his breath rather hard,” related to the respiratory problems involved in the systemic symptoms of Parkinson’s disease [7]. Although there were early observations regarding the pulmonary aspects associated with Parkinson’s disease, actually, the scientific world still is uncertain about all the aspects of the disease. The patterns of ventilatory dysfunction associated with the disease are still unclear and the obstructive patterns, restrictive patterns, respiratory muscle weakness, and sleep breathing disorders (all observed symptoms) also have not had, until now, any certain definition or physiopathological correlations [8].

Regarding the semiology and symptomatology of pulmonary dysfunction, the following symptoms associated with parts of the respiratory system have been observed: coughing or the impairment of coughing, dyspnea, pneumonia, aspiration, exercise intolerance, speech modifications, hypophonia, atelectasis, hypoxia, hypercapnia, sleep apnea and subsequent excessive diurnal somnolence, acute respiratory failure, and difficulty extubating. The medication-induced pulmonary dysfunction can be temporary or permanent. Respiratory problems have been observed and reported later in disease progression related to the fact that patients decrease their level of activity and have a sedentary lifestyle and therefore rarely encounter any respiratory distress [9]. 

Lower airway obstruction is the main characteristic of obstructive lung diseases and a reduction in lung volume is the main characteristic of restrictive lung diseases, with both pulmonary and extrapulmonary etiologies. Extrapulmonary disease usually involves a decrease in lung volume and expansion [5].

Related to speaking and swallowing impairments encountered in Parkinson’s disease, the complex actions of normal speaking and swallowing imply precise synchronization with certain upper airway movements. In the case of upper airway obstruction, speaking and swallowing difficulties follow as a consequence and also present various related modifications such as hypophonia, sleep disordered breathing with increased somnolence during the day, acute respiratory failure, and difficulty extubating. In some cases, if the upper airway musculature is affected, there is also an impairment of the airflow, which can be objectively revealed through obstructive spirometry [10]. The relationships between the symptoms are complex, as can be observed in Figure 1.

In a study by Lee et al. on the clinical pulmonary symptomatology of Parkinson’s disease patients, they reported that 35.8% of patients experienced shortness of breath on exertion, 17.9% reported a cough, and 13% reported production of sputum [11].

Our study selected articles from PubMed and Google Scholar using appropriate search terms. Relevant publications in English from 1950 to 2021 were found by searching using the terms “Parkinson’s disease”, “Parkinson”, and “Parkinson disease” combined with “respiratory”, “pulmonary”, “lungs”, “pulmonary dysfunction”, “respiratory dysfunction”, and “ventilatory dysfunction”. Another search combined the terms “Parkinson’s disease”, “Parkinson”, and “Parkinson disease” and the terms “sleep”, “sleep apnea”, “speech”, “dyspnea”, “swallowing”, and “levodopa respiratory”. Exclusion criteria included animal studies and other neurological disorders different from Parkinson’s disease. The articles obtained from the search were studied, and the relevant matter was analyzed and is described in this paper in the form of a narrative review.

## 2. Neuroanatomy and Pathophysiology of Respiratory Dysfunction in Parkinson’s Disease

The depletion of dopaminergic neurons located in the substantia nigra is the hallmark of Parkinson’s disease and is also followed by considerable destruction of neurons in other locations in the brain such as the brainstem nuclei that control sleep and respiration [12]. Considering the evolution of the morphology of this neurodegenerative disease, the accumulation of alpha-synuclein firstly includes the medulla oblongata in the caudal part and also the respiratory centers which participate in respiratory coordination and the identification of peripheral hypoxemia or hypercapnia [3,13,14,15,16]. The two main characteristics of the disease are dopaminergic depletion and a loss of astrocytic cells which cause impairments in respiratory physiology or at least abnormal awareness of breathing or dyspnea due to a decrease in the production of ATP in key areas of the respiratory centers [17,18,19].

The results of studies in the literature are often contradictory. For example, in a study by Onodera et al. [20], the authors observed that even from the early stages of the disease, patients presented a decreased central vascular response with an insufficient concentration of oxygen and hypercapnia, whereas in a study by Seccombe et al. [21], the authors observed that patients had abnormal respiratory responses to carbon dioxide rather than mild hypoxia and did not present abnormal lung volumes and flows.

## 3. Chest Wall Volume and Asynchrony

Expansion of the rib cage and abdomen involves a synchronized contraction of the diaphragm and the abdominal and intercostal muscles. The prevalence of restrictive respiratory disease in Parkinson’s disease patients causing unsynchronized contraction of the respiratory muscles affecting breathing pattern and chest wall volumes has been found to vary between 28 and 94%, not taking into account the patients’ pulmonary symptoms [9,22,23,24]. The restrictive respiratory pattern in Parkinson’s disease is yet to be entirely understood considering the association with abnormal action regarding the accessory respiratory muscles, abnormal ventilatory control, increased chest wall rigidity, and decreased lung volume due to kyphoscoliosis that are encountered in Parkinson’s disease [25]. In addition, the adverse effects of ergot-derived therapy have been discovered to be at least a risk factor, if not a causative factor, for the restrictive pattern of respiratory function in Parkinson’s disease [22,23].

Florêncio et al. compared 27 Parkinson’s disease patients with healthy subjects and confirmed the presence of restrictive respiratory disease with reduction in the volume of the pulmonary rib cage compartment, as well as observed that half of the Parkinson’s disease patients exhibited paradoxical breathing [26]. Furthermore, different levels of respiratory muscle dysfunction have been observed in patients with Parkinson’s disease. In the same study by Florêncio et al., the Parkinson’s disease patients obtained significantly lower absolute values for FVC and FEV1 as compared with the healthy controls. Regarding respiratory muscle strength, the Parkinson’s disease patients obtained lower scores for maximal inspiratory pressure and maximal expiratory pressure than the control group [26,27]. Nonetheless, an improvement regarding the rigidity of the thoracic wall significantly improved kinesics and coordination at a precise level. A meta-analysis of the literature demonstrated that non-medical treatment methods in Parkinson’s disease provide statistically significant amelioration in respiratory strength measures and peak expiratory flow rates [28]. In a systematic review, Rodriguez et al. concluded that respiratory muscle training might be helpful for enhancing the strength of the respiratory musculature, swallowing, and phonation capacity [29].

## 4. Obstruction 

Several studies have reported the preponderance of upper airway obstruction among Parkinson’s disease patients to be between 6.7 and 67% [22,23], resulting in chronic airflow limitation and, consequently, weakness, bradykinesia, or dysfunctional contraction of the striated musculature in the region of the upper airway [30], which have been considered to be major factors contributing to the development of secretion retention, atelectasis, and aspiration pneumonia. Several studies involving investigative instruments and methods, including spirometry and optic endoscopy, have confirmed the presence of upper airway obstruction [22,23,30,31] and have reported the following two primary changes in the respiratory pattern: respiratory flutter and sudden changes in airflow with an irregular pattern dependent on the glottic and supraglottic structures. Reports from fiberoptic endoscopic examinations have indicated that dysfunctional movements are also present in the regions of the glottis and above the glottis level, which affect the opening and closing of the airway [32]. Furthermore, the tonic activation of vocal fold adductors has been observed and is believed to be connected to a decrease in laryngeal diameter [33]. Changes in the phonatory morphologic components were found to be electromyographically similar to changes experienced in the motor system, including tremor, bradykinesia, and rigidity [34,35]. Correlations between upper airway obstruction and the cardinal characteristics of Parkinson’s disease, i.e., tremor [36], bradykinesia [23], and dystonia [1] have been reported in several studies, raising the possibility of a connection between upper airway obstruction and peripheral motor dysfunctions primarily caused by the effect on the basal ganglia. As a result of the stated dysfunctionalities, respiratory function tests related to upper airway obstruction have revealed a decreased peak expiratory flow rate and maximal flow at 50% of the forced vital capacity, with an increased ratio of forced expiratory volume in one second to the peak expiratory flow rate [37]. Dyspnea, but also hypophonia, pitch changes in the voice, stridor, and wheeze are symptoms that directly follow the upper airway obstruction present in Parkinson’s disease patients [38]. Lower airway obstruction has been reported and associated with rigidity, resistance to passive mobilization, and arthrosis of the cervical spine, which are consistently prevalent in Parkinson’s disease patients, especially in the later stages of the disease [23].

## 5. Dyspnea

Dyspnea has been defined by the American Thoracic Society as “a subjective experience of breathing discomfort that consists of qualitatively distinct sensations that vary in intensity [it] derives from interactions among multiple physiological, psychological, social, and environmental factors, and may induce secondary physiological and behavioral responses” [37].

Dyspnea represents a prevalent symptom in the evolution of Parkinson’s disease, with approximately 40% prevalence in the symptomatology of patients as reported in a study by Baille et al. [39]. The presence of dyspnea correlates with the severity of the disease, decreased ventilatory function, motor fluctuations, dysphagia [39] and even neuropsychological symptoms such as anxiety and depression. The actual mechanism that determines the presence of dyspnea in Parkinson’s disease patients has not yet been discovered but has been associated with the following factors: upper airway obstruction, restrictive respiratory change, levodopa-induced dyskinesia, and hyperventilation. Moreover, studies have reported different data related to the frequency of dyspnea in Parkinson’s disease. Witjas et al. analyzed 50 Parkinson’s disease patients and reported 40% prevalence of dyspnea [40], while the PRIAMO study with a cohort of 1072 patients reported only 1.5% prevalence of dyspnea [41]. 

Since dyspnea can decrease the quality of patients’ lives and is also identified with a reduction in autonomy in ambulatory elderly patients, more studies on dyspnea in Parkinson’s disease patients are required to reach a clearer conclusion. 

Dyspnea and the subsequent ventilatory dysfunctions are considered to be symptoms of autonomic dysfunction. The obstructive and restrictive aspects, the potential drug effects (levodopa can determine diaphragmatic dyskinesias), and an abnormal central control of ventilation are partly responsible for the ventilatory dysfunction in Parkinson’s disease patients.

## 6. Implications of Respiratory Characteristics in the Speech of Parkinson’s Disease Patients

The characteristics of Parkinson’s disease patients’ voices and respiration can have an important impact on the quality of life of these patients. Voice impairments are prevalent in 60–80% of Parkinson’s disease patients and include a monopitched voice, monoloudness, a decrease in the intensity of the voice, a decrease in the pitch of the voice, and a harsh, breathy voice. Patients with Parkinson’s disease present two models of respiratory patterns. The first model is represented by higher lung volumes at the onset and end of speaking as compared with normal lung volumes [42] and the second model is represented by lower lung volumes at the onset and end of speaking as compared with normal lung volumes [43]. The patterns described amplify the requirement of using active muscle forces. The higher lung volumes at the onset and end of speaking as compared with normal lung volumes heighten the necessity of using active inspiratory muscle forces to moderate the descent of the rib cage when encountering high passive recoil forces. The lower lung volumes at the onset and end of speaking as compared with normal lung volumes almost completely depend on active expiratory muscle forces due to the fact that the expiratory muscles represent the main origin of pressure production for speech at low lung volumes. Too much dependence on active muscle forces for speech is unhelpful for Parkinson’s disease patients, not exclusively because it advances the activity of speech breathing, but also because decrements in respiratory muscle strength and control as a consequence of Parkinson’s disease might lead to complications in maintaining and adjusting pressure during speech.

A study by Stegemöller et al. [44] compared the effects of high- and low-dose singing interventions in Parkinson’s disease patients. The results surprisingly showed significant improvements in respiratory outcome measures rather than in all vocal outcome measures.

A previous study by Di Benedetto et al. also showed significant improvements in respiratory measures for Parkinson’s disease patients who participated in singing therapy. These findings are important due to the fact that increased respiratory control may enable more forceful elimination of external material located in the lungs in the case of aspiration pneumonia which represents one of the main causes of death in patients with Parkinson’s disease [45]. 

Furthermore, impairments in speaking and breathing can lead directly to several of the main characteristics of speech disorder in Parkinson’s disease patients, including decreased vocal intensity and brief utterances. The investigative evidence has shown that ongoing intervention procedures that are concentrated on enhancing certain speech inadequacies enhance both speech and breathing. Actually, concentrating on particular speech impairments without precisely impacting speech and breathing might lead to dysfunctional respiratory management and performance [46].

Enhancing the function of expiratory muscles represents an objective for respiratory rehabilitation as it forms a foundation for the thorax and the diaphragm conferring tolerance to sudden powerful contractions at the time of inspiration and aiding in adjusting the pressure for speech. Enhancing expiratory muscle strength in Parkinson’s disease patients can benefit speech breathing and functional speech results by enhancing pressure generation and control. Expiratory muscle strength rehabilitation has been demonstrated to help in cough generation [47] and swallowing capacity in Parkinson’s disease patients [48].

## 7. Implications of Respiratory Characteristics in Swallowing and Aspiration of Parkinson’s Disease Patients

Dysphagia was first reported by James Parkinson as associated with sialorrhea. Dysphagia is frequently correlated with significant clinical problems that are specifically related to a decrease in quality of life, which include inadequate medication consumption, malnutrition, dehydration, aspiration, and, consequently, pneumonia, factors that lead to mortality in patients with Parkinson’s disease. 

Parkinson’s disease-related dysphagia may be caused by a decrease in basal ganglia dopamine activity or other neurotransmitters, but the peripheral mechanisms of the disease are caused by neuromuscular impairments that also subsequently include alterations in pharyngeal muscles related to atrophic myofibers [49]. 

Studies have reported that 80% of patients with Parkinson’s disease develop dysphagia [49]. Usually, swallowing impairment appears at least one year after disease onset, while patients realize the symptoms of dysphagia after approximately ten years; the typical survival period after the presentation of subjective dysphagia is 1–2 years [49,50]. Marano et al. observed that dysphagia is prevalent in approximatively 10% of patients in the first year after diagnosis and that this prevalence doubles in the first three years after the diagnosis; these patients are also more inclined to present non-motor symptomatology (sleep–wake cycle impairments and excessive daytime sleepiness [50]) as compared with Parkinson’s disease patients who do not present this impairment [51]. Parkinson’s disease patients with dysphagia exhibit dysfunctions in all stages of swallowing. In the oral stage, swallowing impairments are defined by drooling, piecemeal deglutition, inadequate mastication, and defective bolus and lingual action management [52]. Impairments in the pharyngeal stage are represented by a lag and a reduction in the hyolaryngeal excursion, a decrease in pharyngeal peristalsis, a lag in the initiation of the swallowing reflex, pharyngeal residue, and aspiration [49,53]. Aspiration is present in approximately half of Parkinson’s disease patients. Studies have described silent penetration and aspiration in Parkinson’s disease patients [54]. Essentially, aspiration leads to an elevated risk of aspiration pneumonia, which is a principal cause of mortality in Parkinson’s disease patients [55]. 

Gaeckle et al. demonstrated a lag in the beginning of the pharyngeal stage of swallowing and also a decrease in hyolaryngeal elevation, which are both predictors of penetration aspiration in Parkinson’s disease patients. The study also revealed that the frequency of penetration and aspiration was directly proportional to the volume of the liquid bolus and the number of times swallowing was performed [56].

## 8. Sleep Disorder Breathing and Daytime Somnolence for Parkinson’s Disease Patients

Non-motor symptoms are prevalent in Parkinson’s disease patients and it has also been recognized that non-motor symptoms play an important negative role in the overall functioning and quality of life of Parkinson’s disease patients. The most representative symptoms for this category are sleep disturbances, daytime sleepiness, cognitive dysfunction, and mood disturbances [12]. 

Sleep disturbances are non-motor symptoms of patients with Parkinson’s disease and include sleep fragmentation, insomnia, restless legs syndrome, acute respiratory failure, and REM sleep behavior disorder.

Recent studies have indicated that obstructive sleep is a potentially important comorbidity in Parkinson’s disease. One study reported that its prevalence is variable and could not conclude that its prevalence was higher than in the general population [57]. A couple of physiopathological processes, i.e., oxidative stress and inflammation, are common in the development of both sleep apnea and Parkinson’s disease; therefore, it is possible that there is a connection between the diseases [58,59]. The most plausible reason for sleep apnea in Parkinson’s disease is the central dysfunction of the respiratory structures from the brainstem in addition to a peripheral airway implication. Furthermore, the data from these findings support the possibility that upper airway motor dysfunction may also be involved, but the most enlightening information comes from the discovery that obstructive sleep apnea could become an aggravating factor for the non-motor symptoms of Parkinson’s disease, especially considering that in the general population, obstructive sleep apnea has been correlated with sleepiness, cognitive and psychomotor dysfunctions [60]. Sleep apnea has been categorized into three types which include central sleep apnea with inadequate activation of the respiratory muscles, obstructive sleep apnea which involves obstruction of airflow despite accurate activation and effort of the respiratory muscles, and a mixed type of sleep apnea representing a combination of the other two types [61].

The main treatment for preventing all the abovementioned symptoms is standard positive airway pressure therapy; however, it results in a variable and incomplete response in obstructive sleep apnea patients [62]. In addition, it has been reported in the literature that Parkinson’s disease is correlated with obstructive sleep apnea, increased sleepiness, and decreased global cognitive function [63]. In a 12-month study by Kaminska et al., standard positive airway pressure therapy in Parkinson’s disease patients with obstructive sleep apnea showed beneficial results in non-motor symptoms globally, subjective sleep quality, cognitive function, and anxiety [64].

## 9. The Effects of Treatment of Parkinson’s Disease on Respiratory Function

An effective antiparkinsonian treatment for the respiratory system has not yet been entirely elucidated. Levodopa treatment is able to moderately enhance the quality of breathing by ameliorating the obstructive and restrictive pulmonary impairments, but without impact or even a negative consequence for ventilatory capacity due to generating diaphragmatic dyskinesias. Tambasco et al. demonstrated that levodopa treatment enhanced the respiratory function of patients and, subsequently, respiratory discomfort was decreased [65]. There were no associations between levodopa and respiratory improvements found in the primary stages of the disease, but in the evolutive stages, medication could account for sustaining the maximum inspiratory oral pressure and sniff nasal inspiratory pressure [66] and could have a beneficial effect on the restrictive pattern of the respiratory impairments by aiding the coordination rather than improving the strength [66]. Paradoxically, antiparkinsonian treatment might produce dyspnea. Dyspnea might appear as a subsequent symptom of the antiparkinsonian treatment, but the underlying cause could be represented by the pleuropulmonary fibrosis caused by treatment with bromocriptine [67] or by diaphragmatic dyskinesias as adverse effects induced by treatment with levodopa [68,69,70]. Improvements in upper airway obstruction due to dopamine treatment have been further highlighted by the acute respiratory events that tend to happen when the therapy is stopped [71,72], for example, neuroleptic malignant-like syndrome, and also sustained by the beneficial response to apomorphine [73,74].

## 10. Conclusions

Parkinson’s disease is commonly correlated with pulmonary impairments. From the initial stages of the disease, respiratory characteristics are considered to be and have been investigated as representing components of the main neurodegenerative disease instead of a distinct disorder. Accordingly, the existence of respiratory symptomatology should be considered by physicians as representing Parkinson’s disease in progress or not well-controlled. Although the role of antiparkinsonian treatment in the respiratory function of these patients is still questionable, the treatment should be considered to present a possible role for improving pulmonary activity, including a possible detrimental contribution to muscle incoordination and exacerbation of shortness of breath in patients with dyskinesias. 

Although the relationships involving the etiology, functionality, physiopathology, and treatment regarding respiratory problems and Parkinson’s disease are not entirely elucidated, the impact on patient’s lives is significant. Consequently, the recognition and prevention of respiratory impairments are clinically valuable.

## Figures and Tables

**Figure 1 brainsci-11-00595-f001:**
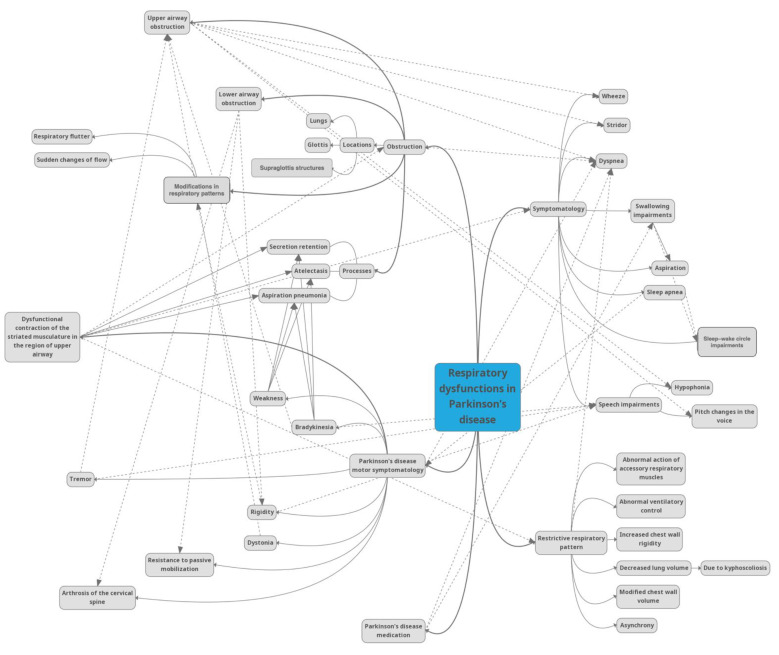
The most significant characteristics in Parkinson’s disease respiratory impairments and the relationships between them.

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
