# Peer review of "Respiratory Dysfunctions in Parkinson’s Disease Patients"

_brainsci, 2021, doi:10.3390/brainsci11050595_

Round 1
Reviewer 1 Report
A method of choosing papers is not described.
Chapters swallowing and aspiration both represent complication of disphagy, they could be unified. The line 155 “” and line 166-167 “” seem repetetive. Therefore, in the last sentence, there’s not any quote of this data.
A complete assessment of the available literature should be implemented, e.g. some dysphagia associated non-motor features have been recently identified among large cohorts (see https://doi.org/10.1016/j.jns.2019.116626)
Other:
Line 24 Substitute “represent” with “represents”
Line 43 “due the” with “due to the”
Line 53-58: “The pattern….psicopathologically”: The sentence is not clear. You could chance it like that: “The pattern of ventilatory dysfunction associated with the disease is still unclear and also the obstructive patterns, restrictive patterns, respiratory muscle weakness and sleep breathing disorders ( all observed symptoms) have not, until now, any certain definition or physiopathologically correlation (8).”
67 – 68:”lower airway..airflow” The sentence is not clear.
72-77: “related… intubated”: The sentence is not clear
77-78: “In some…spirometry” could be change it in: “In some cases, important affection of upper airway musculature may impair airflow and reveal obstructive spirometry”88 connection139 patient’s
Author Response
We have modified the English language and style and also modifications have been made by MDPI Language Editing Service, hopefully will be more approaching and relevant, thank you for your advice and guidance!
A method of choosing papers is not described.
The method is now described.
(lines: 19-22 “Here, we performed a narrative review of the literature and reviewed studies on dyspnea, lung volumes, respiratory muscle function, sleep breathing disorders, and subsidiary speech and swallow impairments related to pulmonary dysfunction in patients with Parkinson’s disease.”)
and lines: 75-85: „Our study selected articles from PubMed and Google Scholar using appropriate search terms. Relevant publications in English, from 1950 to 2021, were found by searching using the following terms: “Parkinson’s disease”, “Parkinson”, “Parkinson disease” combined with “respiratory”, “pulmonary”, “lungs”, “pulmonary dysfunction”, “respiratory dysfunction”, and “ventilatory dysfunction”. Another search combined the terms “Parkinson’s disease”, “Parkinson”, “Parkinson disease” and the terms “sleep”, “sleep apnea”, “speech”, “dyspnea”, “swallowing”, and “levodopa respiratory”. Ex-clusion criteria included animal studies and other neurological disorders different from Parkinson’s disease. The articles obtained from the search were studied, and relevant matter was analyzed and is described in this paper in the form of a narrative review.”)
Chapters swallowing and aspiration both represent complication of disphagy, they could be unified.
Done. We have unified the chapters.
The line 155 “” and line 166-167 “” seem repetetive.
We have modified the text by deleting line 155 to avoid the repetition.
Therefore, in the last sentence, there’s not any quote of this data.
We have added the references.
A complete assessment of the available literature should be implemented, e.g. some dysphagia associated non-motor features have been recently identified among large cohorts (see https://doi.org/10.1016/j.jns.2019.116626)
We have analysed the article and we have included the most interesting findings from it in our study lines 151-156 and another study from the main author as well. Thank you!
Other:
Line 24 Substitute “represent” with “represents”
Done
Line 43 “due the” with “due to the”
Done.
Line 53-58: “The pattern….psicopathologically”: The sentence is not clear. You could chance it like that: “The pattern of ventilatory dysfunction associated with the disease is still unclear and also the obstructive patterns, restrictive patterns, respiratory muscle weakness and sleep breathing disorders ( all observed symptoms) have not, until now, any certain definition or physiopathologically correlation (8).
Done (lines 31-34)
67 – 68:”lower airway..airflow” The sentence is not clear.
It was modified, lines: 44-48.
72-77: “related… intubated”: The sentence is not clear
The paragraph was modified, lines: 65-68.
77-78: “In some…spirometry” could be change it in: “In some cases, important affection of upper airway musculature may impair airflow and reveal obstructive spirometry”
The phrase was modified, lines: 68-73.
88 connection139 patient’s
Done.
Reviewer 2 Report
Dear Editor,
The paper focused on respiratory dysfunction in patients suffered from Parkinson’s disease. The authors collected a wide range of symptoms from chest wall volume and asynchrony to speech, swalloving and daytime somnolence.
The topic is interesting but there are significant concerns.
Major concerns:
- The authors did not presented any aims in the abstract/introduction to describe this review. Eventually, this manuscript seems to be a narrative review but it is not specified.
- I don’t understand what methodology strategy was used for this work because several article, specifically investigating pulmonary dysfunction, have not been mentioned. Es.: Baille G et al (J Parkinsons Dis. 2019;9(4):785-791), Tambasco N et al (J Neural Transm. 2018 Jul;125(7):1033-1036)
- The paper is not well organized: the authors widely described several aspects depending on respiratory dysfunctions such as chest wall volume and asynchrony, obstruction and dyspnea while other aspects are depending on other dysfunctions such as speech, swalloving and daytime somnolence, that should described separately.
- The paper needs of an extensive editing of English language and style.
Author Response
Thank you very much for your help and guidance!
Dear Editor,
The paper focused on respiratory dysfunction in patients suffered from Parkinson’s disease. The authors collected a wide range of symptoms from chest wall volume and asynchrony to speech, swalloving and daytime somnolence.
The topic is interesting but there are significant concerns.
Major concerns:
- The authors did not presented any aims in the abstract/introduction to describe this review. Eventually, this manuscript seems to be a narrative review but it is not specified.
The method is now described.
(lines 19-22 “Here, we performed a narrative review of the literature and reviewed studies on dyspnea, lung volumes, respiratory muscle function, sleep breathing disorders, and subsidiary speech and swallow impairments related to pulmonary dysfunction in patients with Parkinson’s disease.”)
and (lines:60-69 “Our study selected articles from PubMed and Google Scholar using appropriate search terms. Relevant publications in English, from 1950 to 2021, were found by searching using the following terms: “Parkinson’s disease”, “Parkinson”, “Parkinson disease” combined with “respiratory”, “pulmonary”, “lungs”, “pulmonary dysfunction”, “respiratory dysfunction”, and “ventilatory dysfunction”. Another search combined the terms “Parkinson’s disease”, “Parkinson”, “Parkinson disease” and the terms “sleep”, “sleep apnea”, “speech”, “dyspnea”, “swallowing”, and “levodopa respiratory”. Ex-clusion criteria included animal studies and other neurological disorders different from Parkinson’s disease. The articles obtained from the search were studied, and relevant matter was analyzed and is described in this paper in the form of a narrative review.”)
- I don’t understand what methodology strategy was used for this work because several article, specifically investigating pulmonary dysfunction, have not been mentioned. Es.: Baille G et al (J Parkinsons Dis. 2019;9(4):785-791), Tambasco N et al (J Neural Transm. 2018 Jul;125(7):1033-1036)
We have included updated and upgraded the search strategy and included the articles that the reviewer has suggested.
Lines 190-197: “Dyspnea represents a prevalent symptom in the evolution of Parkinson’s disease, with approximately 40% prevalence in the symptomatology of patients as reported in a study by Baille et al. [40]. The presence of dyspnea correlates with the severity of the disease, decreased ventilatory function, motor fluctuations, dysphagia [40] and even neuropsychological symptoms such as anxiety and depression. The actual mechanism that determines the presence of dyspnea in Parkinson’s disease patients has not yet been discovered but has been associated with the following factors: upper airway obstruction, restrictive respiratory change, levodopa-induced dyskinesia, and hyperventilation.”
And lines 263-277: “Tambasco et al. demonstrated that levodopa treatment enhanced the respiratory function of patients and, subsequently, respiratory discomfort was decreased [67]. There were no associations between levodopa and respiratory improvements found in the primary stages of the disease, but in the evolutive stages, medication could account for sustaining the maximal inspiratory oral pressure and sniff nasal inspiratory pressure [68] and could have a beneficial effect on the restrictive pattern of the respiratory impairments by aiding the coordination rather than improving the strength [68]. Paradoxically, antiparkinsonian treatment might produce dyspnea.
- The paper is not well organized: the authors widely described several aspects depending on respiratory dysfunctions such as chest wall volume and asynchrony, obstruction and dyspnea while other aspects are depending on other dysfunctions such as speech, swalloving and daytime somnolence, that should described separately.
We have modified the paper and have added new chapters, hopefully it will be a beneficial upgrade.
- The paper needs of an extensive editing of English language and style.
We have modified the English language and style and also modifications have been made by MDPI Language Editing Service, hopefully will be more approaching and relevant, thank you for your advice and guidance!
Round 2
Reviewer 1 Report
Raised issues have been solved